# Oral Microbial Dysbiosis Driven by Periodontitis Facilitates Oral Squamous Cell Carcinoma Progression

**DOI:** 10.3390/cancers17132181

**Published:** 2025-06-28

**Authors:** Qing Yuan, Hao Wu, Hanyue Tan, Xinxing Wang, Yang Cao, Gang Chen

**Affiliations:** 1Department of Oral and Maxillofacial Surgery, School and Hospital of Stomatology, Tianjin Medical University, 12 Qi Xiang Tai Road, Heping District, Tianjin 300070, China; yq9902132023@163.com (Q.Y.); culiubaoduer@163.com (H.W.); tanhanyue666@163.com (H.T.); 2Department of Environmental Medicine, Military Medical Sciences Academy, 1 Dali Road, Heping District, Tianjin 300070, China; wxxemail@sina.cn

**Keywords:** oral squamous cell carcinoma, periodontitis, microbiota

## Abstract

Oral cancer is a serious disease with poor outcomes, and its causes are still being explored. Periodontal disease is a common and long-lasting inflammation of the tissues around the teeth. Recent research suggests that it may increase the risk of oral cancer. In this study, we used a mouse model to investigate how periodontal disease affects the growth of oral tumors. We found that periodontitis made tumors grow faster and more aggressively. Changes in the bacteria in the oral and tumor area were also observed, along with stronger inflammatory responses in both tissues. These results suggest that periodontitis may promote oral cancer through changes in bacteria and inflammation. This research highlights the importance of oral health, not only for protecting teeth but also for potentially lowering the risk of cancer.

## 1. Introduction

Oral squamous cell carcinoma (OSCC) accounts for over 90% of all oral malignancies and remains a major global contributor to cancer-related mortality [1,2]. Characterized by aggressive local invasion, frequent recurrence, and a high propensity for cervical lymph node metastasis, OSCC poses substantial clinical and therapeutic challenges [3,4]. Despite notable advances in early detection strategies, surgical interventions, radiotherapy, and chemotherapy, the overall prognosis for OSCC patients remains poor, with the 5-year survival rate persistently below 50% [5,6,7]. Established risk factors for OSCC include tobacco use, excessive alcohol consumption, betel quid chewing, human papillomavirus (HPV) infection, and inadequate oral hygiene [8,9].

Periodontitis is a prevalent chronic inflammatory disease initiated by pathogenic microorganisms and closely associated with risk factors such as tobacco use, diabetes, genetic susceptibility, immune dysfunction, and poor oral hygiene [10]. The characteristic features of periodontitis include progressive destruction of the periodontal ligament, the loss of connective tissue attachment, and alveolar bone resorption, ultimately resulting in tooth loss [11]. Growing evidence indicates that dysbiosis of the oral microbiota plays a central role in both the initiation and progression of periodontitis [12,13]. Under healthy conditions, the oral microbiota exists in a state of ecological balance, contributing to the maintenance of immune homeostasis [14,15]. However, periodontitis is marked by a microbial shift toward anaerobic bacterial predominance, a transition that not only drives local periodontal tissue destruction but has also been implicated in a range of systemic diseases, including diabetes mellitus, cardiovascular disease, rheumatoid arthritis, and stroke [16,17,18,19].

Epidemiological evidence has revealed a significant association between periodontitis and OSCC. A large case–control study demonstrated that each millimeter of alveolar bone loss was independently correlated with an increased risk of OSCC, even among non-smokers and non-drinkers, suggesting a direct pathological link between periodontal inflammation and carcinogenesis [20]. Komlós et al. reported that the incidence of oral cancer was 57.1% among patients with periodontitis, compared to 28.6% in those without periodontitis. Moreover, approximately 72.1% of oral cancer patients were diagnosed with stage IV periodontitis [21].

In recent years, increasing attention has been directed toward the potential association between periodontitis and OSCC development [22,23,24]. Sustained chronic inflammation, microbial dysbiosis, and immune modulation have been proposed as key mediators linking these two pathological processes [25,26,27]. In periodontitis, the disruption of microbial homeostasis leads to the overgrowth of anaerobic and pathogenic bacteria, which can produce a range of virulence factors, including lipopolysaccharides, proteases, and metabolic byproducts [28,29,30]. These microbial components not only perpetuate local inflammation and tissue destruction but may also exert systemic effects by modulating immune responses, promoting epithelial–mesenchymal transition (EMT), and enhancing pro-tumorigenic signaling pathways [31,32]. The precise mechanisms through which periodontitis-associated microbial alterations influence the progression of OSCC remain largely unclear. Further research is essential to clarify the direct impact of periodontitis on OSCC and to identify the specific microbial and inflammatory pathways that mediate this relationship.

In this study, we established a periodontitis model combined with OSCC to investigate the relationship between periodontitis and OSCC progression. We hypothesized that periodontitis-induced oral microbial dysbiosis promotes the progression of oral squamous cell carcinoma by enhancing tumor cell proliferation and invasion through inflammatory and microbiota-associated mechanisms.

## 2. Materials and Methods

### 2.1. In Vivo Research Methodology

C57BL/6J mice (male, 4–6 weeks old) were obtained from Beijing Weitong Lihua Biotechnology Co., Ltd. (Beijing, China) and maintained in a specific pathogen-free (SPF) facility. The mice were randomly divided into 2 groups (*n* = 10 per group). Experimental group received periodontitis induction, and the control group received no treatment. Fourteen days later, 5 mice per group were sacrificed to collect periodontal tissues for observing whether the periodontitis model had been successfully established. The remaining 5 mice per group continued to undergo the establishment of a transplanted tumor model. After 10 days, the remaining 5 mice per group were sacrificed, and the serum, periodontal, and tumor tissues were collected. To clarify the experimental timeline, periodontitis was induced over 14 days (Day 0–14), followed by subcutaneous injection of SCC-7 cells on Day 14, and sacrifice on Day 24 (Figure 1A). The study protocol was approved by the Experimental Animal Ethics Committee of Tianjin Institute of Environmental and Operational Medicine (04-2024-020) (Tianjin, China). The overall experimental design and timeline are summarized in Figure 1A, providing a schematic overview including periodontitis induction, tumor inoculation, and tissue collection.

### 2.2. Establishment of Experimental Periodontitis Model

PD group animals were anesthetized by intraperitoneal injection of 300 μL 0.025 mg/mL Tribromoethanol. Following immobilization, the operative field was exposed. A stainless steel ligature wire was horizontally inserted at the left maxillary first molar, looped around the cervical region of the tooth, and tightly secured before trimming the excess wire. Mice were maintained on normal diet with daily monitoring of ligature status.

### 2.3. Establishment of OSCC Model

OSCC was established in mice via subcutaneous injection of SCC-7 cells into the right dorsal flank. Prior to injection, the target area was depilated. A suspension of SCC-7 cells (5 × 10^6^ cells/mL, containing 1 × 10^6^ cells in 200 μL) was administered subcutaneously and gently massaged to ensure absorption. All mice were sacrificed ten days post-inoculation. Tumor volume and weight were then measured, with volume calculated as (length × width^2^)/2.

### 2.4. Micro-CT Analysis

Micro-computed tomography (micro-CT) was employed to measure bone mineral density (BMD), trabecular number (Tb.N), and trabecular thickness (Tb.Th). Following successful establishment of the periodontitis model, maxillary samples were collected and scanned using a SkyScan 1276 scanner (Bruker, Karlsruhe, Germany) with an image pixel size of approximately 10 µm at 85 kVp and 200 µA. Three-dimensional images were reconstructed using NRecon (Version 1.7.3.1), generating approximately 2000 tomographic slices per sample. Two-dimensional micro-CT images were acquired using DataViewer (Version 1.5.4.6) to visualize bone morphology and relative grayscale intensity of bone density. Quantitative analyses of BMD, Tb.N, and Tb.Th were performed using CTAn (Version 1.18.8.0). All measurements and data analyses were conducted in triplicate, with mean values calculated for final results.

### 2.5. Histopathological Analysis of Maxillae and OSCC Tissues

Following micro-CT scanning, the isolated maxillary specimens were first fixed in 4% paraformaldehyde solution, then decalcified in 14% EDTA solution (pH 8.0) at room temperature for 4–5 weeks. After standard dehydration procedures, the samples were embedded in paraffin using standard techniques. Tissue sections of 4 μm thickness were prepared along the mesiodistal plane using a microtome, followed by hematoxylin and eosin (H&E) staining for histological observation and analysis. For tumor tissue samples, after dehydration and embedding, the embedded tissues were sectioned into 4–6 μm thick slices using a microtome. The tissue sections were stained with hematoxylin and eosin (H&E) solution. Following dehydration and clearing treatments, the sections were mounted for microscopic observation, and data analysis was performed using CaseViewer software (Version 2.4.0.119028).

### 2.6. Quantitative Real-Time PCR Analysis of Periodontal and OSCC Tissues

A 20 µL reaction system was prepared with 4 µL master mix, 16 µL of DEPC-treated ddH_2_O, and RNA added based on concentration. Reverse transcription was performed at 37 °C for 15 min (3 cycles), followed by 85 °C for 5 sec and 4 °C hold. For qPCR, each 20 µL reaction contained 1 µL cDNA, 0.5 µL of each forward/reverse primer, 8 µL of DEPC-treated ddH_2_O, and 10 µL of SYBR GREEN MIX. GAPDH served as the reference gene (ΔCt method), with relative expression calculated using 2^−ΔΔCt^. Primer sequences are shown in Table 1.

### 2.7. In Vitro Culture and Scratch Assay of SCC-7 Cells

In total, 1.5 mL of pre-warmed trypsin was added to digest SCC-7 cells for 3 min. The cell suspension was collected and centrifuged for 4 min. After cell counting, 4000 cells/well were seeded in a 96-well plate with 100 μL complete medium (6 replicates/group). Following overnight culture, cells were treated with mouse serum from Con and PD groups for 48 h. Prepare CCK-8 working solution at a ratio of serum-free medium to CCK-8 solution = 10:1; then, incubate for 2 h. Absorbance at 450 nm was measured using a microplate reader to calculate proliferation rates.

SCC-7 cells were cultured in 6-well plates for 48 h. At 90% confluency, scratch wounds were created in each well. The NC group received normal medium, while the other two groups were treated with mouse serum from Con and PD groups. Images were captured at 0 h and 48 h post-scratch. Scratch areas were measured and quantified using ImageJ software (Version 1.54p).

### 2.8. Microbiome Sampling and 16S rDNA Gene Amplicon Sequencing

16S rDNA analysis of oral and tumor microbiota was conducted. DNA was extracted from mouse oral and tumor tissues using CTAB method. The V4 hypervariable region of 16S rDNA was amplified using 515F/806R primers (Sangon Biotech, Shanghai, China). Library concentration was measured by Qubit fluorometry and qPCR, with fragment size analyzed by bioanalyzer. Libraries were pooled proportionally and sequenced on Illumina platforms. Data analysis included Pielou’s evenness, Chao1 richness, and top 10 microbial compositions at phylum and class levels.

### 2.9. Statistical Analysis and Methods

All experiments were repeated ≥3 times and analyzed using SPSS 26.0. Continuous data are presented as mean ± SD after normality testing. After verifying parametric assumptions, intergroup comparisons used independent samples *t*-tests (α = 0.05).

## 3. Results

### 3.1. Murine Periodontitis Model Successfully Established

The schematic diagram outlines the experimental workflow used to establish the periodontitis and OSCC animal models (Figure 1A). The experimental setup ensured stable ligature placement in the PD group without compromising oral mucosal health (Figure 1B). The Con group maintained intact gingival epithelium and normal alveolar bone height, whereas the PD group exhibited characteristic periodontitis-associated changes, including the disruption of epithelial integrity at the dento-gingival junction, the loss of periodontal ligament attachment, and pronounced inflammatory cell infiltration (Figure 1C). Micro-CT analysis demonstrated significant alveolar bone resorption and decreased bone density in the PD group (Figure 1D,E), accompanied by substantial reductions in BMD, Tb.N and Tb.Th (Figure 1F–H). These pathological and imaging findings accurately recapitulate key features of human periodontitis, confirming the successful induction of experimental periodontitis in this mouse model.

**Figure 1 cancers-17-02181-f001:**
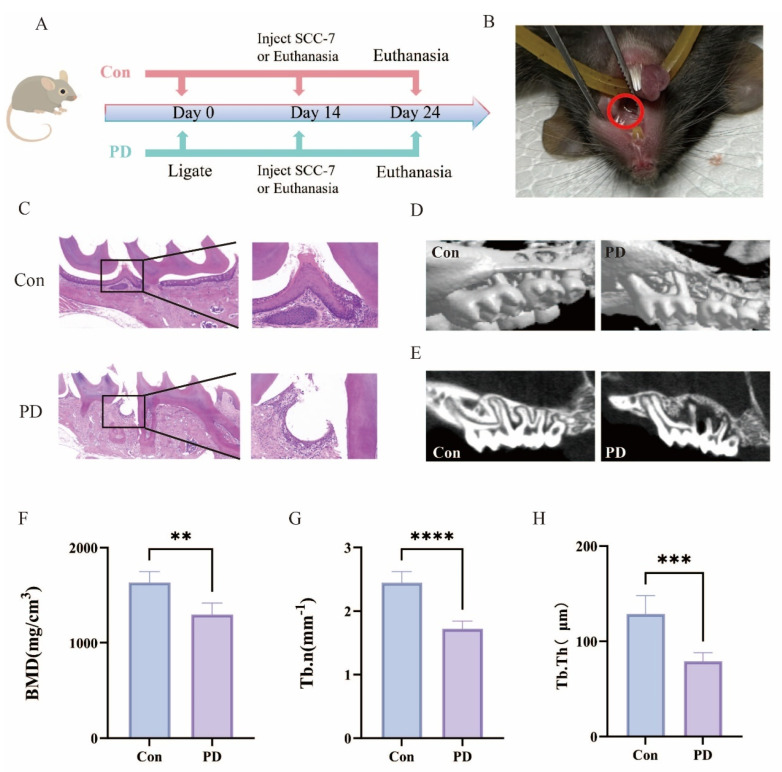
Successful establishment of the periodontitis model: (**A**) Schematic diagram of the experimental design. (**B**) Stainless steel ligatures successfully placed around the teeth, indicated by the red circles. (**C**) Representative HE-stained tissue sections, with the right panels showing magnified views of the boxed regions on the left. (**D**) Representative 3D micro-CT images. (**E**) Representative sagittal 2D micro-CT images. (**F**) Quantitative analysis of BMD (** *p* < 0.01). (**G**) Analysis of Tb.N (**** *p* < 0.0001); (**H**) Analysis of Tb.Th (*** *p* < 0.001).

### 3.2. Periodontitis Exhibits a Significant Positive Association with OSCC Progression

To investigate the relationship between periodontitis and the progression of OSCC, the SCC-7 model was successfully established in both the Con and PD groups (Figure 2A). Comparative analysis revealed that OSCC in the PD group exhibited significantly greater volume and weight compared to the Con group (Figure 2B,C). Molecular analysis further showed an upward trend in the expression of the proliferation marker Ki67 in the PD group, as assessed by qPCR (Figure 2D). Consistently, H&E staining revealed enhanced tumor vascular proliferation and extensive areas of sheet-like necrosis in the PD group (Figure 2E). These findings suggest a positive correlation between periodontitis and the growth and progression of OSCC.

### 3.3. Periodontitis Enhances Proliferation and Migration of SCC-7 Cells

The wound healing assays demonstrated significantly enhanced migratory capacity of SCC-7 cells when cultured with serum from the PD group compared to Con, as evidenced by greater wound closure area and increased relative migration rates after 48 h of incubation (Figure 3A,B). CCK-8 proliferation assays revealed that the PD group serum significantly promoted SCC-7 cell growth relative to the Con serum during the same experimental period (Figure 3C). These results collectively indicate that periodontitis-derived serum factors possess tumor-promoting properties capable of stimulating both migratory and proliferative activities in OSCC cells, likely through the action of inflammatory mediators present in the circulation. The observed effects suggest a potential mechanistic link between periodontal inflammation and the enhanced oncogenic behavior of malignant oral epithelial cells.

### 3.4. Periodontitis Alters Microbial Abundance in Oral and OSCC

To assess microbial diversity in oral and OSCC tissues between Con and PD groups, we conducted Chao1 richness and Pielou’s evenness (pielou_e) analyses. The Pielou_e index revealed a significant increase in oral from the PD group compared to controls (Figure 4A), whereas no significant difference was observed in OSCC tissues (Figure 4B). Similarly, the Chao1 richness index demonstrated a moderate elevation in the oral cavity of the PD group and an elevation in OSCC tissues from the PD group relative to controls (Figure 4C,D). These findings suggest an upward trend in microbial richness and evenness associated with periodontitis in both oral and OSCC environments.

### 3.5. Periodontitis Alters Microbial Communities in Oral and OSCC Tissues

Given the well-established association between periodontitis and changes in the oral microbiota, we further employed 16S rDNA sequencing technology to analyze the microbial communities in oral and OSCC tissues. We conducted a taxonomic analysis at the phylum level and identified the top ten phyla in terms of abundance. In the oral cavity, compared with the Con group, the abundances of *Proteobacteria* and *Bacteroidota* in the PD group increased, while the abundances of *Firmicutes* and *Actinobacteriota* decreased (Figure 5A). A similar pattern was also observed in the OSCC tissues, where the abundances of *Proteobacteria* and *Bacteroidota* increased and those of *Firmicutes* and *Actinobacteriota* decreased in the periodontitis group (Figure 5B). At the class level, analysis revealed increased abundances of *Clostridia, Bacteroidia*, and *Gammaproteobacteria* and decreased abundances of *Bacilli*, *Verrucomicrobiae*, and *Alphaproteobacteria* in the oral cavity of the PD group versus the Con group (Figure 5C). A similar pattern was also observed in the OSCC tissues, with more significant increases in the abundances of *Bacteroidia* and *Gammaproteobacteria* (Figure 5D). These results indicate that periodontitis significantly alters the composition of the oral microbiota. Notably, taxonomic groups that constitute the resident oral microbiota were also detected in the OSCC tissues, and the changes in OSCC microorganisms were consistent with those observed in the oral cavity. This implies that periodontitis not only causes local changes in the microbial community but also has systemic effects and may potentially regulate the OSCC microenvironment.

### 3.6. Upregulation of Cytokines in Periodontal and OSCC Tissues

The mRNA expression levels of IL-1β, IL-10, and IL-6 were significantly elevated in periodontal tissues of the PD group compared to controls, while no significant differences were observed for IL-18 and TNF-α (Figure 6A–E). Similarly, OSCC tissues from the PD group showed increased mRNA levels of IL-1β, IL-10, TNF-α, and IL-18, although IL-6 expression was not significantly altered (Figure 6F–J). These results indicate a consistent pro-inflammatory trend in both periodontal and OSCC tissues under periodontitis conditions.

## 4. Discussion

Periodontitis is a highly prevalent chronic inflammatory disease of the oral cavity, marked by the progressive destruction of the periodontal ligament, the resorption of alveolar bone, and ultimately, tooth loss [10,11,33]. In this study, we successfully established a ligature-induced periodontitis murine model. By integrating the periodontitis model with the OSCC model, we demonstrated that the presence of periodontitis significantly accelerated OSCC progression. Furthermore, serum derived from periodontitis mice exhibited a direct proliferative effect on SCC-7 cells, suggesting that, beyond local tissue damage, periodontitis may exert broader pro-tumorigenic effects through circulating inflammatory mediators and other systemic factors [34,35].

Pathogenic bacteria in the oral cavity are recognized as the primary initiating factor in the development of periodontitis [36,37,38]. Certain keystone pathogens, such as *Porphyromonas gingivalis*, *Treponema denticola*, and *Tannerella forsythia*, are particularly instrumental in disrupting host–microbial homeostasis, provoking chronic inflammatory responses, and driving the progressive destruction of periodontal tissues [37,39,40,41]. Emerging evidence increasingly supports the notion that periodontal pathogens also contribute to the progression of various malignancies [27,29,42]. These bacteria exert their oncogenic influence primarily through the modulation of inflammatory signaling pathways, the promotion of EMT, and the facilitation of immune evasion mechanisms [43,44,45]. *P. gingivalis* has been implicated in the pathogenesis and advancement of pancreatic and esophageal cancers, while it has shown a positive association with the occurrence and progression of breast cancer [46,47,48,49,50,51]. Collectively, these observations underscore the broad systemic effects exerted by periodontal pathogens and position oral microbial dysbiosis as a critical mechanistic link not only in the destruction of periodontal tissues but also in the development and progression of cancer.

Microbial profiling revealed that periodontitis induces marked alterations in the oral and OSCC tissue microbiota, especially evident at the phylum and class levels. We observed increased abundances of *Proteobacteria* and, accompanied by reductions in *Firmicutes* and *Actinobacteriota*, both in oral and OSCC tissues of the PD group. *Bacteroidota* includes notable periodontal pathogens such as *P. gingivalis* and *Prevotella intermedia*, both of which have been implicated in promoting tumorigenic processes through the modulation of inflammatory responses, the activation of matrix metalloproteinases, and the enhancement of EMT [37,52,53,54]. Elevated *Bacteroidia* abundance, particularly driven by *P. gingivalis*, has been linked to pancreatic cancer and esophageal adenocarcinoma, suggesting a conserved oncogenic role across tissue types [37,55]. Similarly, *Proteobacteria*, which encompass genera such as *Escherichia*, *Salmonella*, and *Pseudomonas*, have been shown to contribute to cancer-related inflammation, genotoxicity, and metabolic dysregulation [56,57,58]. Their enrichment in both oral and OSCC tissues raises the possibility that periodontitis-associated microbial shifts may create a pro-tumorigenic environment through shared inflammatory and mutagenic mechanisms.

Conversely, *Firmicutes* and *Actinobacteriota*, typically dominant in healthy oral microbiota, are often associated with protective roles, including the maintenance of mucosal barrier integrity and the suppression of pathogenic overgrowth [59,60]. The decline of beneficial *Bacilli*, a class within *Firmicutes*, and *Actinobacteriota* in periodontitis may thus reflect a loss of microbial-mediated homeostatic functions, further amplifying the inflammatory and oncogenic potential of the dysbiotic community [61].

At the class level, data showed elevated levels of *Clostridia*, *Bacteroidia*, and *Gammaproteobacteria*, alongside reductions in *Bacilli*, *Verrucomicrobiae*, and *Alphaproteobacteria* in the PD group. *Clostridia* include species that can shift from beneficial to pathogenic roles, with some producing toxins or secondary bile acids implicated in cancer-related inflammation and DNA damage [62]. The enrichment of *Bacteroidia*, which contains key periodontal pathogens like *Porphyromonas* and *Prevotella*, has been linked to enhanced inflammation, epithelial invasion, and immune evasion in oral and gastrointestinal cancers [63,64]. Emerging evidence indicates that *Gammaproteobacteria*, encompassing opportunistic pathogens like *Escherichia coli* and *Pseudomonas aeruginosa*, are capable of producing genotoxic compounds and reactive oxygen species that directly induce DNA damage and promote tumorigenesis [65,66]. More alarmingly, when intestinal microbiota homeostasis is disrupted, these pathogenic bacteria may translocate through the gut-oral axis [67]. Clinical observations reveal that gut-derived *Gammaproteobacteria* not only form resilient biofilms on oral mucosa, but their secreted genotoxins also synergize with indigenous oral pathogens to collectively compromise epithelial barrier integrity. This pathogenic synergy perpetuates the activation of inflammatory pathways and ultimately creates a tumor-promoting microenvironment conducive to oral carcinogenesis [68,69]. Together, these class-level shifts highlight the profound microbial reorganization in periodontitis and its potential influence on OSCC progression, underscoring the importance of further dissecting these microbial contributions to cancer biology [70].

Within the transplanted tumors, the expression levels of IL-1β and IL-6 were significantly elevated, mirroring the changes observed in the periodontal tissues. This suggests that periodontitis may promote OSCC development by enhancing the release of pro-inflammatory cytokines. IL-1β, a key mediator in the pro-inflammatory cytokine network, promotes osteoblast apoptosis and enhances osteoclast activity, thereby accelerating alveolar bone resorption [71,72]. TNF-α can activate the NF-κβ signaling pathway, inducing inflammatory cell infiltration and exacerbating periodontal tissue destruction [71]. The marked increase in IL-6 expression indicates that periodontitis may influence distant tissues through systemic inflammatory responses. IL-6 not only facilitates neutrophil recruitment during the acute phase of inflammation but also sustains a pro-inflammatory microenvironment during chronic inflammation via the JAK/STAT3 signaling pathway [73,74]. The upregulation of these cytokines not only reflects the severity of local inflammation but may also have broader implications for host systemic health. This supports a pathogenic axis of periodontitis–microbial dysbiosis–cytokine induction–tumor microenvironment remodeling.

Several limitations of this study should be taken into consideration. The OSCC model was established at an ectopic site spatially separated from the oral cavity, resulting in the lack of significant differences in Pielou and Chao1 diversity indices observed between tumor tissues from the Con and PD groups [75,76,77,78]. Additionally, the reliance on a single murine OSCC cell line may limit the generalizability of our findings, as it does not fully encompass the molecular and phenotypic heterogeneity characteristic of human oral squamous cell carcinoma [79,80,81]. Although the ligature-induced periodontitis model is extensively utilized and well-established for mimicking periodontal inflammation in mice, it cannot entirely recapitulate the complex microbial ecology and immunological landscape of human periodontitis [82]. Consequently, caution is advised when extrapolating these preclinical findings to clinical scenarios. To strengthen translational relevance, further studies involving human clinical specimens and longitudinal analyses are imperative to corroborate the mechanistic insights derived from this model.

In summary, this study reveals that periodontitis-associated microbial dysbiosis and the accompanying dysregulation of cytokines collectively shape a tumor-promoting microenvironment. These alterations not only drive periodontal destruction but also facilitate OSCC progression. Targeting the oral microbiota–cytokine axis could represent a promising strategy for oral cancer prevention and intervention.

## Figures and Tables

**Figure 2 cancers-17-02181-f002:**
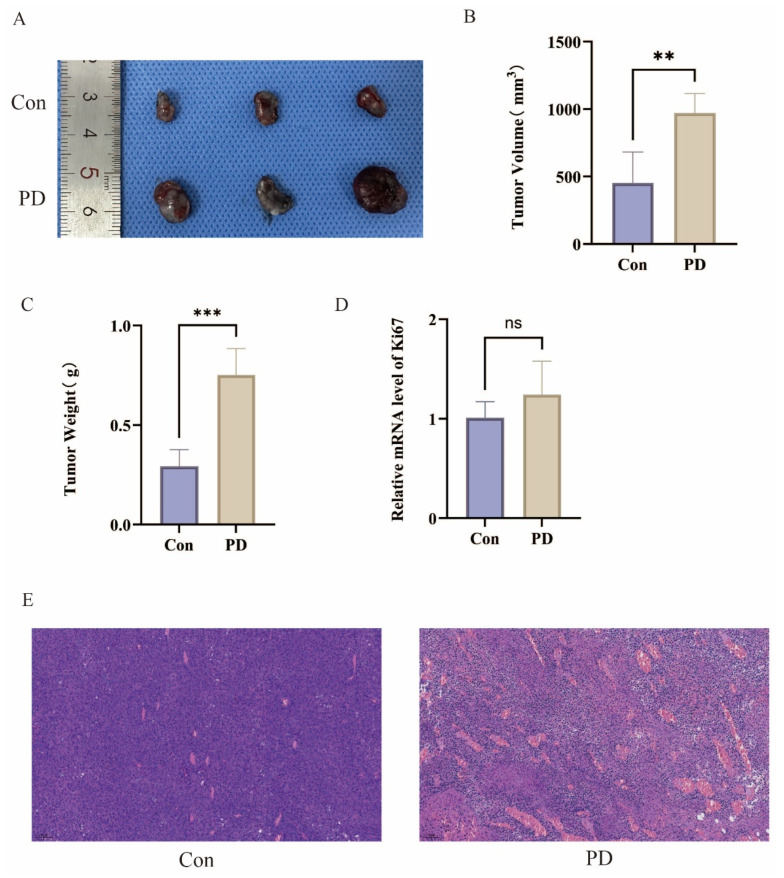
Periodontitis promotes the progression of OSCC: (**A**) Representative images of OSCC. (**B**) Analysis of differences in OSCC volume (** *p* < 0.01). (**C**) Analysis of differences in OSCC weight (*** *p* < 0.001). (**D**) Ki67 mRNA expression levels (ns, not significant). (**E**) Representative HE-stained tissue sections.

**Figure 3 cancers-17-02181-f003:**
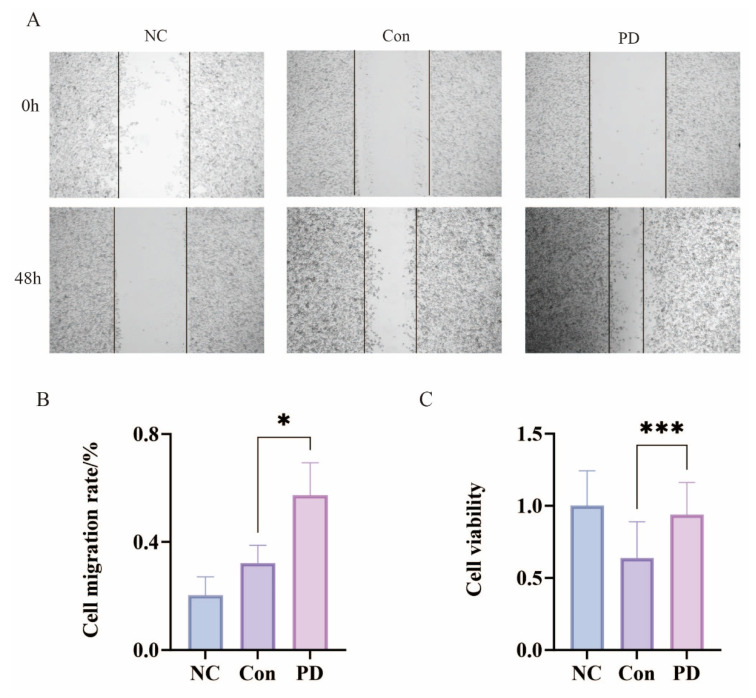
Periodontitis promotes proliferation and migration of SCC-7 cells: (**A**) Wound healing assay to assess the migration ability of SCC-7 cells. (**B**) Bar graph showing changes in wound area (* *p* < 0.05). (**C**) Relative proliferation rate of SCC-7 cells (*** *p* < 0.001).

**Figure 4 cancers-17-02181-f004:**
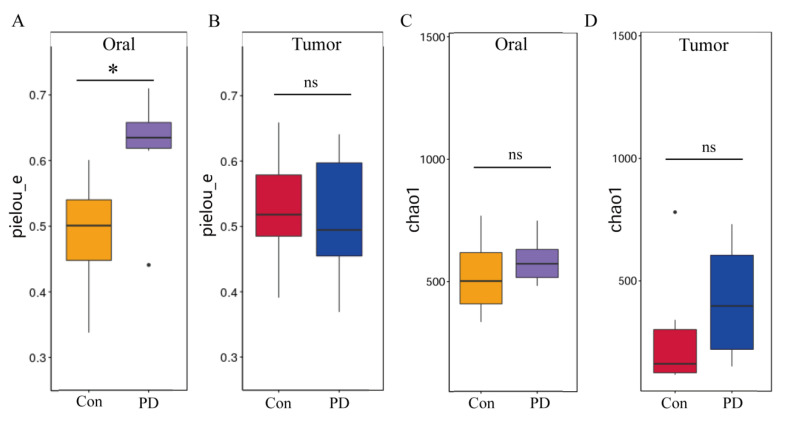
Periodontitis increases microbial richness and evenness in oral and OSCC tissues: (**A**) pielou_e richness analysis of oral microbiota (* *p* < 0.05); (**B**) pielou_e richness analysis of OSCC microbiota; (**C**) Chao1 richness analysis of oral microbiota; (**D**) Chao1 richness analysis of OSCC microbiota. ns, not significant.

**Figure 5 cancers-17-02181-f005:**
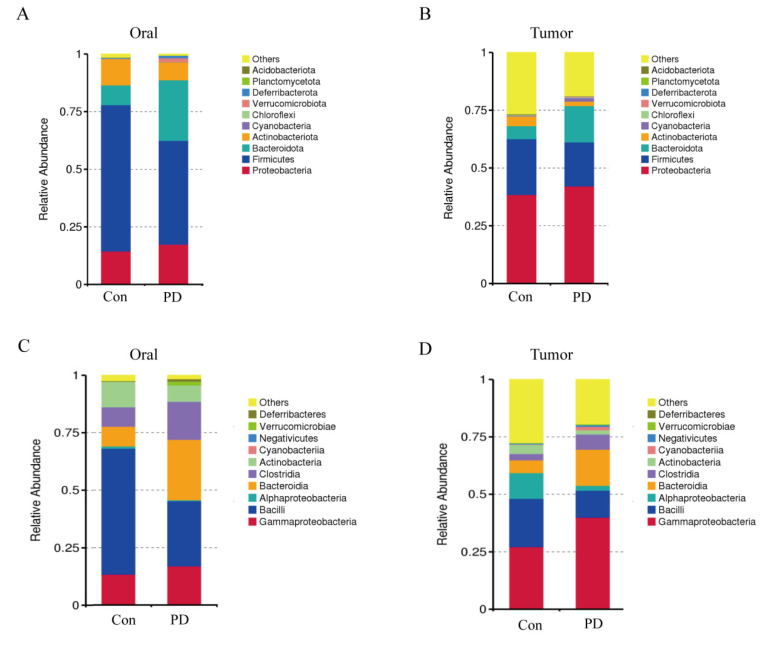
Top 10 bar plots show that periodontitis induces similar microbial shifts in oral and OSCC tissues: (**A**) Top 10 bar plot of oral microbiota at the phylum level. (**B**) Top 10 bar plot of OSCC microbiota at the phylum level. (**C**) Top 10 bar plot of oral microbiota at the class level. (**D**) Top 10 bar plot of OSCC microbiota at the class level.

**Figure 6 cancers-17-02181-f006:**
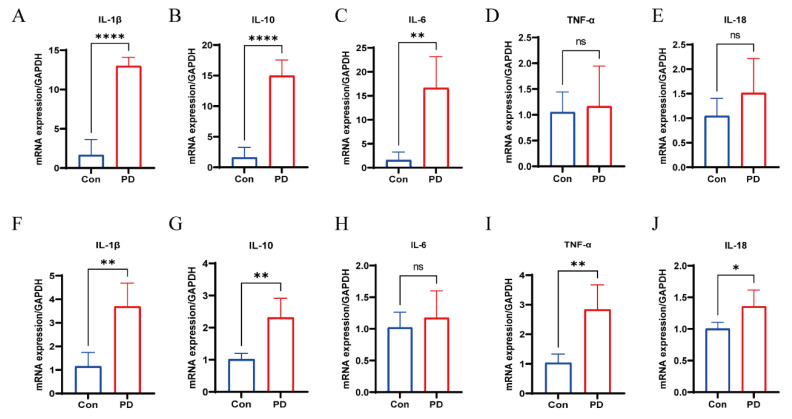
Periodontitis increased cytokine mRNA levels: (**A**–**E**) mRNA expression levels of IL-1β, IL-10, IL-6, TNF-α, and IL-18 normalized to GAPDH in periodontal; (**F**–**J**) mRNA expression levels of IL-1β, IL-10, IL-6, TNF-α, and IL-18 normalized to GAPDH in OSCC. * *p* < 0.05, ** *p* < 0.01, **** *p* < 0.0001, ns, not significant.

**Table 1 cancers-17-02181-t001:** Primer sequences of qRT-PCR.

Primer	5′→3′
Ki67	F:GTCCATAAAGACCCTTTTCAGCCA
	R:ATGTCCTCGTTTCCGATTATAGT
IL-1β	F:CTACAGGCTCCGAGATGAACAAC
	R:TCCATTGAGGTGGAGAGCTTTC
IL-10	F:GGAAGACAATAACTGCACCCACTT
	R:AAGGCAGTCCGCAGCTCTAG
IL-6	F:AGTTGCCTTCTTGGGACTGATG
	R:GGGAGTGGTATCCTCTGTGAAGTCT
TNF-α	F:GGTCCCCAAAGGGATGAGAA
	R:TGAGGGTCTGGGCCATAGAA
IL-18	F:GACAGCCTGTGTTCGAGGATATG
	R:TGTTCTTACAGGAGAGGGTAGAC
GAPDH	F:GGTTGTCTCCTGCGACTTCA
	R:TGGTCCAGGGTTTCTTACTCC

## Data Availability

All data generated or analyzed during this study are included in this published article.

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
