# Peer review of "Oral Microbial Dysbiosis Driven by Periodontitis Facilitates Oral Squamous Cell Carcinoma Progression"

_cancers, 2025, doi:10.3390/cancers17132181_

Round 1
Reviewer 1 Report
Comments and Suggestions for Authors
- Figure 2 showed the enhanced SCC-7 xenograft growth, and the tumor pictures and the tumor measurement should be plotted to exhibit the formed tumors. Further, method sections 2.1 to 2.3 and figures 1-2 were suggested re-arranged, because these experiments were operated continually and sequentially.
- The authors demonstrated the periodontitis facilitating OSCC progression by xenograft, and the environmental factor could be the probable factor to enhance tumor growth. Did they check the cytokine production and/or interleukins from the mice? It may underscore the significance of the periodontitis in mediating the microenvironment and/or tumor niches.
- Considering the tumor heterogenicity and varying individual differences, the additional oral cancer cell line is required to support their observations and theory. Consequently, functional assays in Figure 3 could be performed in another cell line.
- It is confusing that tumor microbiota did not exhibit the difference between control and PD groups (in the figure 4), despite the PD provided a favorable condition for cancer growth. Concurrently, the authors needed to indicate the sites for tumor injection in the method to clarify why the Pielou and Chao1 indicators were similar in tumor sections of two groups.
- Microbiota analyses showed the altered expression pattern by the PD induction. Whether the short-term antibiotics administration could reverse the PD phenotypes? It may reveal the mechanistic interpretation for PD-mediated OSCC progression.
- The authors employed the mouse model to highlight the association of periodontitis to OSCC progression, and it needs the clinical relevance to underpin this viewpoint.
Not assessed
Author Response
请参阅附件。

Reviewer 2 Report
Comments and Suggestions for Authors
Authors performed important research to shed light on relationship between periodontitis and oral squamous cell carcinoma, and to define the potential role of oral microbiota dysbiosis in the development of pro-tumorigenic microenvironment.
Marked alterations induced by periodontitis are described, modulation of the tumor microenvironment is discussed.
Methods are reported in detail, the quality of illustrations is quite good, the procedure of analysis and modelling is explained. The work was carried out at a good scientific level and using modern methods. All necessary sources are cited, the description is logical and competent.
Authors stated that future direction in this field should be the influence of therapeutic agents on studied processes.
The article is interesting to wide audience, and the material can be published after minor revision.
Ref. 7. – art number 3156
Ref.28 – art number 1736
Ref 30. - art number lgz002
Ref. 39 - art number 9915
Ref. 55 - art number 3418
Ref. 65 - art number 2124
Ref. 66 - art number 2271
Reviewer 3 Report
Comments and Suggestions for Authors
This study is exceptionally well-designed and thoroughly executed. The combination of a periodontitis model with SCC-7 xenografts provides a novel and relevant approach to exploring the interplay between periodontal inflammation and the progression of oral squamous cell carcinoma (OSCC).
The approach is comprehensive, addressing both the direct biological effects on tumor proliferation and invasion, as well as shifts in the oral microbiota, using advanced tools such as 16S rDNA sequencing. This depth of analysis adds significant value to the study and supports its conclusions.
The manuscript is clearly written, the results are well interpreted, and the conclusions are logical and well-founded. Overall, this work makes a meaningful contribution to our understanding of how chronic inflammation may influence oral carcinogenesis.
Title: The current title is adequate, but it could be refined to more precisely reflect the main focus of the work—particularly the interaction between microbial dysbiosis and the progression of oral squamous cell carcinoma (OSCC).
Introduction: The introduction provides a solid background and contextualizes the study well. However, it would be helpful to explicitly state the central hypothesis. Doing so would clarify the objective and better guide the reader through the rationale of the experimental design.
Murine Periodontitis Model Successfully Established : Consider clarifying the timeline of disease induction and tumor development in the methods section to help readers better follow the experimental progression.
Figures: The quality of the figures is high, and they support the data effectively. If available, including additional micro-CT images—especially those illustrating bone density changes in the periodontitis model—would enhance the morphological component of the study.
Please include a dedicated paragraph in the discussion section addressing the study’s limitations, such as the use of a single cell line and the animal model's translatability to human disease.
References: The bibliography is well curated, current, and relevant to the topi
